# Soybean Yield Response of Biochar-Regulated Soil Properties and Root Growth Strategy

**Di Wu, Weiming Zhang \*, Liqun Xiu, Yuanyuan Sun, Wenqi Gu, Yuning Wang, Honggui Zhang and Wenfu Chen**

Biochar Engineering & Technology Research Center of Liaoning Province, Agronomy College, Shenyang Agricultural University, Shenyang 110866, China; didi0055@126.com (D.W.); xiuxiu1990@outlook.com (L.X.); 2020200076@stu.syau.edu.cn (Y.S.); wenqi0120@126.com (W.G.); wyuning1204@126.com (Y.W.); honggui321@126.com (H.Z.); wfchen5512@126.com (W.C.)
\* Correspondence: biochar_zwm@syau.edu.cn

**Abstract:** Continuous cropping obstacles limit soybean production. Biochar is beneficial for soybean growth, but it is unclear whether biochar performs a sustainable role in continuous cropping. To further explore the effects of biochar on soil properties and soybean growth, a ceramic Wagner pot-simulated field experiment was conducted with biochar at different doses (12, 24, and 48 t·ha$^{-1}$) over a period of 3 years, no fertilizer, no biochar (CK) and fertilizer (F)treatments were used for comparison. The results showed that biochar significantly reduced soil bulk and improved the rhizosphere soil pH, available nutrients (N, K), and total nutrients (C, N, P, and K) compared with CK and F. Moreover, the soybean root length, surface area, volume, and exudates increased with biochar amendment. In particular, biochar significantly increased the nodule number, dry weight, and nitrogenase activity of soybean. Furthermore, biochar promoted soybean growth and increased soybean yield. In general, we found that the soybean yield increased with biochar and that biochar had a positive, sustainable effect on soil properties and soybean root growth, providing a new cultivation measure for soil health and soybean production in continuous cropping, which is very important for increasing soybean productivity to break the limitations of soybean traditional continuous cultivation.

**Keywords:** soybean; biochar; continuous cropping; soil properties; root

## 1. Introduction

Soybean is an important grain and oil crop worldwide. China's soybean production does not meet the market supply, and a number of factors limit sustainable soybean production. Continuous cropping obstacles have been reported as the main factors limiting the growth and development of soybean [1], and they are closely related to the deterioration of rhizosphere microecological and root growth restriction [1,2]. At present, soybean continuous cropping obstacles are mainly overcome by coating, fertilization, soil sterilization, and changes in cropping systems (e.g., crop rotation) [3,4]. However, these measures cannot fundamentally address the deterioration of soil structure and physicochemical properties under continuous cropping. Therefore, how to fundamentally improve soil structure and characteristics, change the rhizosphere microecological environment so that it is conducive to soybean growth, and promote soil health and sustainable soybean production have become important issues that need to be addressed for sustainable soybean production in China.

In China's agricultural sector, over 50% of straw is burned or discarded, resulting in serious environmental pollution and waste of resources [5]. Biochar, a new research hotspot in recent years, has been widely researched and recognized by scientists around the world. Straw and other agricultural wastes can be carbonized into biochar through pyrolysis reactions and then returned to the field for agricultural applications, enabling efficient recycling of straw and other waste resources [6]. Biochar is alkaline; has good physical and chemical characteristics such as microporosity; a large specific surface area



and strong adsorption capacity; is rich in many nutrient elements; and is a stable, carbon-rich product [7]. Studies have shown that biochar significantly improves soil structure and quality, crop root growth, crop production capacity and yield [8–10]. More notably, biochar has a sustainable effect on soil and crops. Biochar contains a high carbon content and has a stable aromatic carbon structure, which can exist in the soil for a long time and have a long-term impact on the soil structure and physicochemical properties, providing a sustainable beneficial soil environment for crop growth [11]. Many years of biochar improvement experiments have found that biochar can increase the soil porosity, pH value, and soil water holding capacity; enrich soil organic matter; promote soil nitrogen mineralization; and increase yields of crops such as soybean, rice, and sorghum [12,13]. Biochar can also promote soybean growth and development. Studies have shown that biochar can increase soybean yield by enhancing soil nutrient levels (such as the availability of soil phosphorus and some trace elements) [14] and boosting soybean nutrient absorption (such as increasing plant absorption and accumulation of nitrogen, as well as phosphorus and potassium) [15]. Biochar also performs an important role in regulating soybean root growth and the rhizosphere microecological environment. Previous results have shown that the addition of biochar can increase the length, total area, and volume of soybean roots in the early growth stage and promote root growth [16,17]. In addition, biochar application can increase the number and weight of soybean nodules and enhance the physiological functioning of soybean [18]. Other studies have suggested that biochar can significantly inhibit the severity of root rot and promote the healthy growth of soybean roots [19].

In summary, studies have shown that biochar is conducive to promoting soybean growth. However, the effects of biochar application under continuous cropping conditions on the rhizosphere soil environment and soybean growth and development are less well-studied. It is uncertain whether biochar has a positive effect on the sustained growth of soybean yield under continuous cropping conditions. The beneficial effect of biochar has the potential to break through the time and space constraints on soybean production, improve the sustainable yield of soybean, and enable farmers to gain better economic returns. Therefore, we adopted a pot experiment for three years to investigate the sustainable effects of straw biochar on rhizosphere soil physicochemical properties, root growth (root morphogenesis, growth, and physiological function), soybean growth, and yield under continuous cropping from a soil and crop health perspective, determining the appropriate biochar dose for soybean yield increasing under continuous cropping. This research provides new technical measures for promoting continuous soybean cropping and improving soybean planting areas and yield to promote the sustainable development of soybean production, which have important scientific and application value.

## 2. Materials and Methods

### 2.1. Experimental Site

The test soil is classified as a Hapli-Udic Cambisol (FAO Classification). To simulate the field experiment conditions, the ceramic Wagner pot method, which is recognized by international experts, was adopted. Each pot (diameter, 35 cm; height, 40 cm) contained 15 kg soil. Soybean (Tiefeng 40) was provided by the Tieling Academy of Agricultural Sciences. Five treatments were set: no fertilizer, no biochar (CK); fertilizer treatment, no biochar (F); and biochar doses of 12 t·ha$^{-1}$ (80 g/pot) (C1), 24 t·ha$^{-1}$ (160 g/pot) (C2), and 48 t·ha$^{-1}$ (240 g/pot) (C3). Each treatment was set up in 30 pots arranged in a random block design (each block had 10 pots). Soybean was cultivated on a yearly basis, and two soybean plants were planted per pot. The biochar was mixed thoroughly with soil in the pots before planting soybean in 2016, and no biochar application occurred in the following years (2017 and 2018). For the F treatment, the fertilizer (compound fertilizer, NPK 12-18-15) amount was 300 kg/hm$^2$ (5 g/pot), according to the recommendation from the soybean breeder, and was applied each year (2016–2018). Water management, pest control, and other cultivation measures were consistent across all treatments. The soil basic physicochemical properties were 1.46 g/kg total N (TN), 0.58 g/kg total P (TP), 18.35 g/kg total K (TK), 88.00 mg/kg

available N (AN), 18.80 mg/kg available P (AP), 83.50 mg/kg available K (AK), and pH 5.46. The raw biochar material was straw: pyrolysis temperature, 400–450 °C; particle size, 0.30–0.35 cm; pH, 9.24; C, 59.58%; nitrogen content, 0.87%; phosphorus content, 0.66%; and potassium content, 1.02%. The straw was provided by Liaoning Golden Future Agriculture Technology Co., Ltd. (Xiuyan, Liaoning, China).

## 2.2. Soil Property Analysis

At maturity (5 October) in 2016–2018, the soil bulk density was measured using the ring knife method. Soil total N was determined through elemental analyzer (Vario MACRO Cube, Elementar, Germany). Soil total phosphorus (TP) and total potassium (TK) were determined by the NaOH melting method, TP was determined by the molybdenum antimony colorimetric method, and TK was determined by the flame photometer method [18]. Rhizosphere soil samples were collected at the seedling stage (25 June), flowering stage (26 July), and podding stage (20 August). We collected the soybean plant with its roots, gently removed bulk soil, and then collected soil that was attached to the root as much as possible using brushes. Subsequently, the rhizosphere soil was naturally dried indoors, screened through a 2 mm sieve, and stored. Soil pH was measured by compound glass electrode (Ohaus SC310, Auhaus Co., Ltd., Freehold, NJ, USA) with a water–soil ratio of 2.5:1 [20]. The soil available nitrogen (AN) was determined by the alkali diffusion method. The soil available phosphorus (AP) and soil available potassium (AK) were determined by the molybdenum antimony colorimetric method after extraction with $NaHCO_3$ and the flame photometer method after extraction with $CH_3COONH_4$, respectively [20].

## 2.3. Measurement of Soybean

Potted soybean was planted on 23 May 2016, 25 May 2017, and 25 May 2018. From 2016 to 2018, three pots of representative soybean plants were selected randomly from each treatment at the seedling stage, flowering stage, and podding stage. The aboveground stems, leaves, and pods of the plants were separated and placed in an oven, then dried at 80 °C to a constant weight and weighed on an electronic balance. At soybean maturity (8 October), three pots of representative soybean plants were randomly selected from each treatment, and the pod number, pod weight, grain number, grain weights of the stem and branches, and weight of 100 grains were recorded.

Samples were taken at the seedling stage, flowering stage, and podding stage. The aboveground part of the soybean was cut from the cotyledon node. The soil in the basin was separated from the root. The roots were obtained, and nodules were collected by careful and repeated washing. The root was scanned and analyzed by a root scanner (WinRHIZO STD4800 LA2400, Regent Instruments, Quebec, QC, Canada). The total root length (TRL), total root area (TRA), total root volume (TRV), and mean root diameter (MRD) were measured. Cuts were made at cotyledon nodes, and the roots were placed in bags filled with absorbent cotton; the exudates were collected for 12 h and weighed to measure root exudation [21]. The root active absorbing surface area (RAA) was determined using the methylthionine chloride dipping method [21]. Nitrogenase activity in nodules was determined by the acetylene reduction assay (ARA) [22], and the determination conditions of gas chromatography were as follows: column temperature, 60 °C; injection temperature, 120 °C; FID monitoring temperature, 120 °C; gas flow rates, $N_2$ = 50 mL/min, $H_2$ = 60 mL/min, and air = 50 mL/min.

## 2.4. Data Analysis

Microsoft Excel 2019 and SPSS 17.0 software (IBM, Chicago, IL, USA) were used to process and analysis the data. One-way analysis of variance (ANOVA) and Duncan's significant difference analysis were used for variance analysis and multiple comparisons of the treatment and temporal (year) effects ($p < 0.05$; $n = 3$); Graph Pad Prism 5.0 was used for plotting.

## 3. Results

### 3.1. Effects of Biochar on Soil Properties under Soybean Continuous Cropping

3.1.1. Soil Bulk Density

Biochar application significantly reduced the soil bulk density (Figure 1). C3 was 9.76% lower than CK, and C2 and C1 were 6.00% and 4.63% lower than CK, respectively. For three consecutive years, the soil bulk density of C3, C2, and C1 was 7.95%, 4.23%, and 3.09% lower than F, respectively. According to the analysis of interannual variation, C2 and C3 treatments in 2017 and 2018 was significantly lower than that in 2016, and C1 non-significant difference, indicating that biochar with higher application rate has a sustainable impact on soil bulk density.

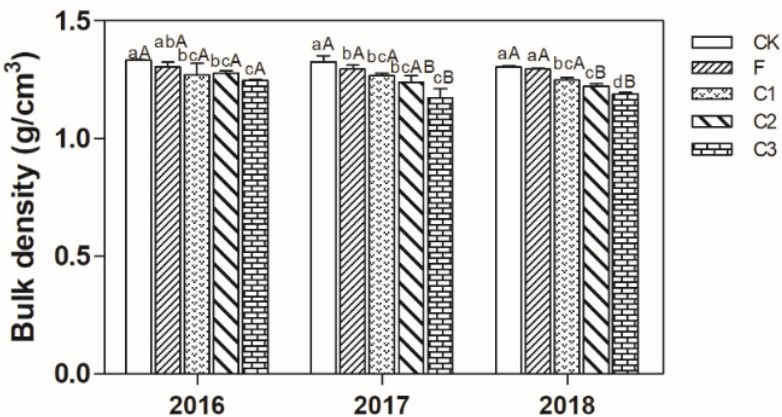

**Figure 1.** Effects of biochar on soil bulk density under soybean continuous cropping. Different lowercase letters indicate differences ($p < 0.05$) between the treatments in each year, and different capital letters indicate differences of the same treatment in 3 years. Bars represent standard errors ($n = 3$).

3.1.2. Soil pH and Nutrients

The soil pH of all biochar treatments was significantly higher than those of the CK and F treatments. C3 increased the pH by 7.23%, 16.46%, and 16.91% compared with CK, and increased the pH by 7.23%, 13.52%, and 10.67% compared with F in 2016–2018. According to the interannual variation, there was no significant difference, indicating that a one-time application of biochar can effectively maintain the pH of rhizosphere soil for continuous soybean cropping (Table 1).

**Table 1.** Effect of biochar on rhizosphere soil pH and macronutrient content under soybean continuous cropping.

| Traits | Year | CK | F | C1 | C2 | C3 |
|---|---|---|---|---|---|---|
| pH | 2016 | 6.65 ± 0.05 [bA] | 6.65 ± 0.22 [abA] | 6.71 ± 0.08 [bB] | 7.30 ± 0.48 [aA] | 7.13 ± 0.31 [abA] |
| | 2017 | 6.56 ± 0.14 [cAB] | 6.73 ± 0.03 [cA] | 7.57 ± 0.07 [abA] | 7.54 ± 0.07 [aA] | 7.64 ± 0.07 [bA] |
| | 2018 | 6.39 ± 0.07 [cB] | 6.75 ± 0.12 [bB] | 7.41 ± 0.07 [aA] | 7.45 ± 0.04 [aA] | 7.47 ± 0.08 [aA] |
| TC | 2016 | 12.87 ± 1.00 [dA] | 12.73 ± 0.38 [dA] | 14.40 ± 0.17 [cA] | 18.65 ± 0.15 [bA] | 24.64 ± 0.28 [aA] |
| | 2017 | 12.20 ± 0.17 [dA] | 11.06 ± 0.26 [eB] | 13.84 ± 0.35 [cA] | 17.42 ± 0.17 [bB] | 23.01 ± 0.16 [aB] |
| | 2018 | 10.67 ± 0.61 [dB] | 11.08 ± 0.05 [dB] | 13.13 ± 0.13 [cB] | 16.83 ± 0.07 [bC] | 19.73 ± 1.04 [aC] |
| TN | 2016 | 1.78 ± 0.05 [cA] | 1.92 ± 0.08 [bA] | 1.76 ± 0.04 [cA] | 1.98 ± 0.04 [bA] | 2.25 ± 0.06 [aA] |
| | 2017 | 1.62 ± 0.02 [cB] | 1.51 ± 0.01 [dB] | 1.49 ± 0.01 [dB] | 1.87 ± 0.07 [bB] | 2.19 ± 0.02 [aA] |
| | 2018 | 1.54 ± 0.06 [cB] | 1.57 ± 0.07 [cB] | 1.79 ± 0.03 [bA] | 1.91 ± 0.03 [aAB] | 1.99 ± 0.03 [aB] |

**Table 1.** *Cont.*

| Traits | Year | CK | F | C1 | C2 | C3 |
|---|---|---|---|---|---|---|
| TP | 2016 | 0.56 ± 0.01 cB | 0.65 ± 0.01 bA | 0.63 ± 0.03 bA | 0.66 ± 0.02 bA | 0.80 ± 0.02 aA |
|  | 2017 | 0.62 ± 0.02 bA | 0.59 ± 0.03 cB | 0.64 ± 0.01 bA | 0.65 ± 0.01 bA | 0.70 ± 0.01 aB |
|  | 2018 | 0.59 ± 0.01 bAB | 0.59 ± 0.01 bB | 0.64 ± 0.01 aA | 0.65 ± 0.00 aA | 0.65 ± 0.01 aC |
| TK | 2016 | 18.12 ± 0.50 cA | 19.19 ± 0.26 bA | 19.91 ± 0.14 aA | 20.11 ± 0.10 aB | 20.09 ± 0.66 aA |
|  | 2017 | 12.70 ± 0.63 cC | 12.63 ± 0.45 cB | 15.07 ± 0.96 bB | 14.46 ± 0.94 bC | 20.67 ± 0.45 aA |
|  | 2018 | 15.56 ± 0.60 dB | 20.35 ± 2.45 cA | 20.83 ± 0.24 bA | 22.30 ± 0.55 aA | 20.57 ± 0.56 bA |
| AN | 2016 | 95.67 ± 4.21 aA | 93.33 ± 3.09 aA | 86.33 ± 6.50 aA | 94.50 ± 4.04 aAB | 87.83 ± 4.06 aA |
|  | 2017 | 76.00 ± 1.15 dB | 87.00 ± 1.09 cA | 89.67 ± 1.45 bcA | 100.00 ± 1.40 aA | 92.00 ± 1.15 bA |
|  | 2018 | 76.00 ± 1.15 cB | 75.33 ± 0.88 cB | 75.67 ± 0.88 cA | 87.67 ± 0.88 bB | 96.33 ± 1.45 aA |
| AP | 2016 | 24.41 ± 2.36 aA | 34.96 ± 1.84 aA | 30.57 ± 6.30 aA | 25.15 ± 2.43 aAB | 26.52 ± 1.54 aB |
|  | 2017 | 18.63 ± 0.27 dAB | 26.67 ± 0.38 cB | 28.70 ± 0.40 bA | 28.90 ± 0.35 bA | 31.20 ± 0.49 aA |
|  | 2018 | 14.70 ± 0.36 dB | 16.83 ± 0.26 cC | 23.60 ± 0.34 aA | 21.83 ± 0.23 bB | 24.03 ± 0.23 aB |
| AK | 2016 | 106.67 ± 4.41 bA | 110.00 ± 10.41 bA | 120.00 ± 5.77 bA | 125.00 ± 13.23 bA | 160.00 ± 5.77 aA |
|  | 2017 | 91.67 ± 0.76 cB | 89.33 ± 0.42 cdA | 87.67 ± 1.99 dB | 127.00 ± 0.55 aA | 123.33 ± 1.33 bB |
|  | 2018 | 88.67 ± 3.18 cA | 109.33 ± 1.45 bA | 102.67 ± 0.88 bA | 122.00 ± 0.58 aA | 127.00 ± 2.89 aA |

The data values are mean ± SD (*n* = 3). Duncan's test was used for multiple comparisons. Different lowercase letters indicate differences (*p* < 0.05) between the treatments in each year, and different capital letters indicate differences of the same treatment in 3 years. (*n* = 3). TN: total nitrogen; TP: total phosphorus; TK: total potassium; AN: available nitrogen; AP: available phosphorus; AK: available potassium.

The soil TC increased with the biochar application rate. The soil TC of the biochar treatments was also significantly higher than that of CK and F. From 2016 to 2018, the soil TC of C3, C2, and C1 was 75.33%, 48.48%, and 16.13% greater per year than that of CK, respectively, and 93.23%, 51.97% and 18.92% greater per year than that of F, respectively. Interannual analysis showed that the soil TC decreased significantly after three years of continuous cropping.

The soil TN increased with the biochar application rate. The soil TN of C3, C2, and C1 was 29.99%, 23.33%, and 2.36% greater per year than that of CK, respectively. The soil TN of the biochar treatments was also significantly higher than that of F. Interannual analysis showed that the soil TN of CK and F decreased significantly after three years of continuous cropping, but it did not change significantly in the biochar treatments.

From 2016 to 2018, compared with that of CK, the TP in the rhizosphere soil of C3, C2, and C1 increased 21.29%, 10.67%, and 7.55%, respectively. In 2017 and 2018, the TP of all biochar treatments was significantly higher than that of F. In 2016, only the TP concentration in the rhizosphere soil of C3 was significantly higher than that of F.

Biochar application increased the TK concentration with the increase in the biochar application rate. Compared with CK, the soil TK of C3, C2, and C1 increased 35.29%, 22.72%, and 13.54%, respectively, per year on average. C3, C2, and C1 showed increases of 25.48%, 11.45%, and 11.01% per year on average compared with F, respectively. The TK of C2 in 2018 was significantly higher than that in the first two years, showing that the application of biochar can effectively improve the soil TK concentration, demonstrating obvious cumulative and sustainable effects.

Biochar application significantly increased AN, AP, and AK in the rhizosphere soil of soybean. In 2017 (2018), the AN content in the rhizosphere soil of C2 was 31.56% (15.36%) and 14.94% (15.86%) higher than that of CK and F, respectively; the AN content in the rhizosphere soil of C3 was 21.05% (6.75%) and 5.75% (27.30%) higher than that of CK and F, respectively. In 2017, the soil AP content of C3 was 67.47% and 16.99% higher than that of CK and F, respectively. In 2018, the soil AP content of C3 was 63.47% and 42.28% higher than that of CK and F, respectively. There was no obvious interannual variation in the AP concentration in the rhizosphere soil under continuous soybean cropping. From 2016 to 2018, the AK content increased with the increase in the biochar application rate in the order

C3 > C2 > C1, and the AK content under biochar application was higher than that under F and CK (Table 1).

### 3.2. Regulation of Biochar on Soybean Roots under Continuous Cropping

#### 3.2.1. Effects of Biochar on Soybean Root Morphology

The application of biochar in the soybean seedling and flowering stages had a decreasing effect on the MRD over the course of plant development. The MRD of CK at the seedling stage was thicker than that of C3 and F, and at the flowering stage was 23.48% thicker than that of C3 (Figure 2A). Biochar significantly increased the TRA, TRV, and TRL of soybean. At the seedling stage, the TRA of C1 was 61.36% and 49.54% higher than that of CK and F, respectively. C3 inhibited root growth in soybean seedlings and that C2 promoted root growth in soybean at flowering and podding (Figure 2B). At the seedling stage, the TRV of the C1, C2, and C3 was significantly higher than CK and F. At the podding stage C2 was 31.91% and 25.77% higher than that of CK and F, respectively, and the TRV of C3 was 28.06% and 22.10% higher than that of CK and F, respectively (Figure 2C). At the flowering and podding stage, the TRL of C2 was significantly higher than that of the other treatments (Figure 2D).

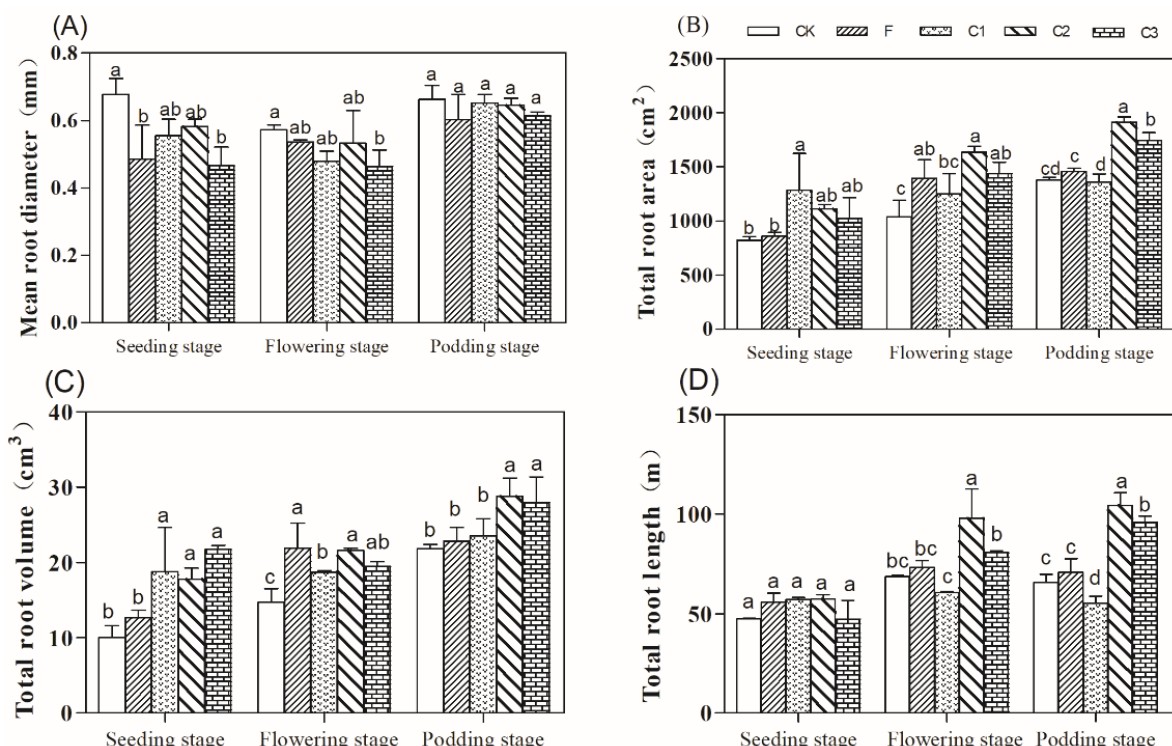

**Figure 2.** Effects of biochar on soybean root morphology under continuous cropping. (**A**) Effect of biochar on soybean mean root diameter under continuous cropping; (**B**) Effect of biochar on soybean total root area under continuous cropping; (**C**) Effect of biochar on soybean total root volume under continuous cropping; (**D**) Effect of biochar on soybean total root length under continuous cropping. Columns with different letters are significantly different at *p* < 0.05. Bars represent standard errors (*n* = 3). TRL: total root length; TRA: total root area; TRV: total root volume; MRD: mean root diameter.

#### 3.2.2. Effect of Biochar on the Physiological Activity of Soybean Roots

Biochar significantly increased soybean root exudation in a biochar application rate-dependent manner. The root exudate amount of C3 was significantly higher than that of the other treatments, with a value 19.82 times and 8.27 times that of CK and F, respectively (Figure 3).

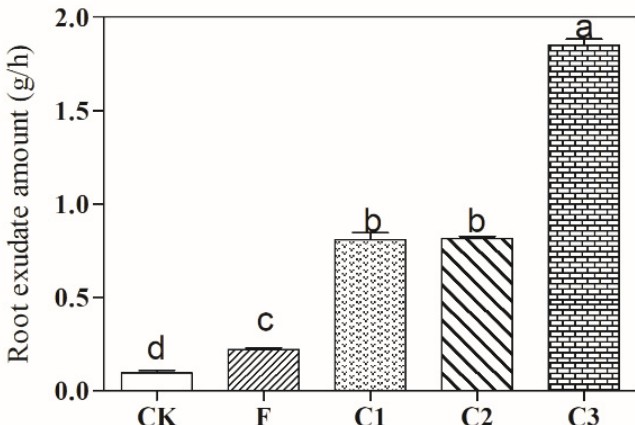

**Figure 3.** Effects of biochar on root exudate amount under soybean continuous cropping. Columns with different letters are significantly different at $p < 0.05$. Bars represent standard errors ($n = 3$).

The RAAs of F and C1 were significantly higher than that of CK at the flowering stage, with values 3.90% and 2.59% higher than that of CK. At the podding stage, the RAA of C2 soybean was 1.60% higher than that of CK. This finding indicated that excessive biochar application (C3) had no significant effect on the RAA, and the appropriate application of biochar is particularly important (Figure 4).

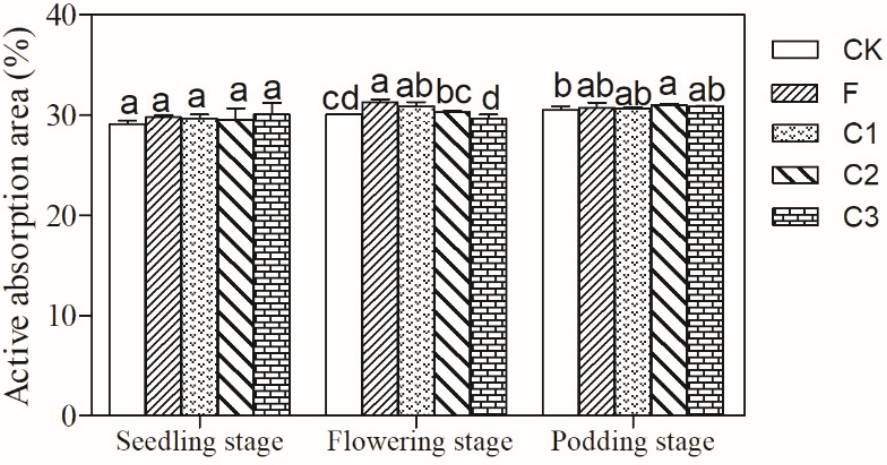

**Figure 4.** Effect of biochar on soybean root active absorption area under continuous cropping. Columns with different letters are significantly different at $p < 0.05$. Bars represent standard errors ($n = 3$). RAA: root active absorbing surface area.

The number and dry weight of soybean nodules in different treatments showed the same trend: C3 had significantly higher values than CK (Figure 5B). The nodule number of C3 was 2.88 times that of CK, and the dry nodule weight of C3 was 2.78 times that of CK (Figure 5A). The results showed that the high application rate of biochar significantly promoted the growth of soybean nodules. The root nodule nitrogenase activity treated with biochar and fertilization was higher than that of the control. The root nodule nitrogenase activities of F, C2, and C1 were 1.71, 1.58, and 1.57 times that of CK, respectively. The root nodule nitrogenase activities of C2 and C1 were higher than those of C3, indicating that the promoting effect under the high biochar application rate was poor (Figure 5C).

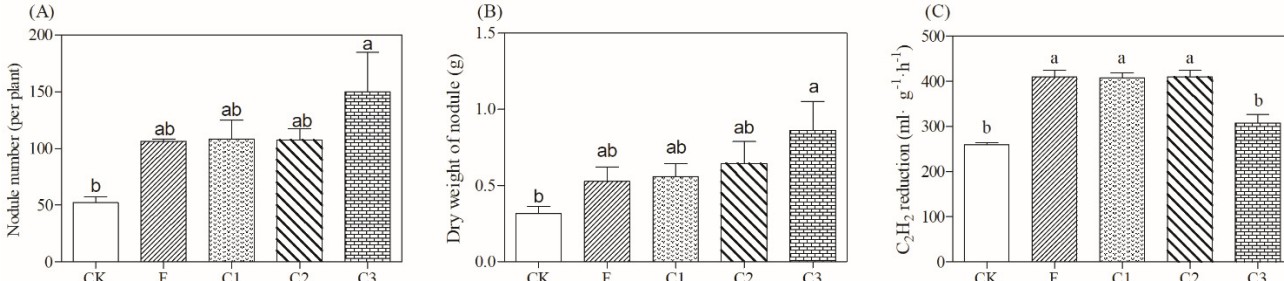

**Figure 5.** Effect of biochar on nodule number (**A**), dry weight (**B**), and nitrogenase activity (**C**) under soybean continuous cropping. Columns with different letters are significantly different at *p* < 0.05. Bars represent standard errors (*n* = 3).

### 3.3. Effects of Biochar on Soybean Growth and Yield under Continuous Cropping

Soybean plant height under each biochar treatment showed the following trend: C2 > C3 > C1, indicating that C2 (24 t·ha$^{-1}$) had the greatest promoting effect on the plant height. Compared with the CK, the average plant height of C2 increased 8.17% every year from 2016 to 2018. In 2016, the soybean aboveground dry matter weight of F and C2 was higher than CK. The trend in biochar treatments was C2 > C3 > C1. In 2017, the biochar treatments had a substantial influence on soybean aboveground dry matter weight, which was related to the continuous output of biochar nutrients. In 2018, C2 had the highest dry matter weight under continuous cropping (Table 2).

**Table 2.** Effects of biochar on soybean growth and yield under continuous cropping.

| Year | Treatment | Plant Hight (cm) | Dry Matter Weight (g) | Pod Number per Plant | Grain Number per Plant | Grain Weight per Plant (g) | 100-Grain Weight (g) |
|---|---|---|---|---|---|---|---|
| 2016 | CK | 65.17 ± 1.88 [bA] | 28.77 ± 0.51 [cC] | 31.00 ± 1.15 [bB] | 49.33 ± 4.67 [cB] | 8.37 ± 0.41 [bB] | 17.11 ± 0.08 [bB] |
| | F | 68.83 ± 0.59 [aA] | 34.13 ± 2.27 [aB] | 43.67 ± 3.18 [aB] | 72.67 ± 4.36 [abB] | 11.70 ± 0.57 [abB] | 18.72 ± 0.20 [aB] |
| | C1 | 63.83 ± 2.93 [bA] | 30.33 ± 2.01 [bcC] | 43.00 ± 5.29 [aB] | 61.00 ± 10.12 [bcC] | 11.30 ± 1.17 [abC] | 19.29 ± 0.40 [aB] |
| | C2 | 69.83 ± 1.38 [aA] | 32.96 ± 1.10 [abC] | 44.00 ± 1.53 [aB] | 77.67 ± 3.38 [aB] | 12.33 ± 1.22 [aB] | 18.77 ± 0.15 [aB] |
| | C3 | 70.33 ± 1.90 [aA] | 32.02 ± 2.35 [abcC] | 46.67 ± 1.45 [aB] | 80.67 ± 1.76 [aB] | 11.67 ± 1.24 [abB] | 18.57 ± 0.07 [aB] |
| 2017 | CK | 65.63 ± 1.40 [bA] | 32.42 ± 2.56 [aB] | 38.00 ± 2.00 [bB] | 67.330 ± 4.67 [aAB] | 14.43 ± 0.43 [bAB] | 17.28 ± 0.54 [cB] |
| | F | 72.47 ± 1.01 [aA] | 36.08 ± 1.12 [aB] | 45.33 ± 2.85 [abB] | 77.00 ± 4.36 [aB] | 15.96 ± 0.74 [abB] | 19.58 ± 0.59 [bB] |
| | C1 | 66.97 ± 2.79 [bA] | 32.71 ± 0.99 [aB] | 56.00 ± 7.81 [aAB] | 89.00 ± 10.12 [aB] | 16.29 ± 0.82 [abB] | 22.14 ± 0.42 [aA] |
| | C2 | 72.43 ± 2.37 [aA] | 35.54 ± 1.70 [aB] | 52.33 ± 2.33 [abB] | 85.67 ± 3.38 [aB] | 17.50 ± 0.29 [aB] | 21.26 ± 0.13 [aA] |
| | C3 | 67.27 ± 1.81 [bA] | 34.56 ± 3.82 [aB] | 49.67 ± 1.20 [abB] | 81.67 ± 1.76 [aB] | 15.51 ± 0.31 [bB] | 22.18 ± 0.61 [aA] |
| 2018 | CK | 62.00 ± 2.65 [bA] | 36.09 ± 2.56 [aA] | 58.00 ± 4.04 [dA] | 100.67 ± 18.11 [bA] | 20.51 ± 3.38 [bA] | 20.11 ± 0.58 [bA] |
| | F | 68.67 ± 3.51 [aA] | 40.42 ± 1.87 [aA] | 97.33 ± 2.03 [aA] | 150.00 ± 7.64 [aA] | 28.37 ± 2.58 [aA] | 22.21 ± 0.35 [aA] |
| | C1 | 61.67 ± 3.51 [bA] | 36.71 ± 0.91 [bA] | 71.67 ± 4.10 [cA] | 119.67 ± 3.53 [abA] | 24.10 ± 1.29 [abA] | 21.26 ± 0.13 [abA] |
| | C2 | 66.33 ± 2.08 [abA] | 38.21 ± 0.91 [abA] | 82.33 ± 3.84 [bA] | 157.33 ± 14.66 [aA] | 29.53 ± 1.95 [aA] | 22.14 ± 0.42 [aA] |
| | C3 | 65.00 ± 2.08 [abA] | 40.34 ± 1.90 [aA] | 82.00 ± 2.65 [bA] | 152.67 ± 3.76 [aA] | 28.48 ± 1.32 [aA] | 22.18 ± 0.61 [aA] |

The data values are mean ± SD (*n* = 3). Duncan's test was used for multiple comparisons. Different lowercase letters indicate differences (*p* < 0.05) between the treatments in each year, and different capital letters indicate differences of the same treatment in 3 years. (*n* = 3).

In continuously cropped soybean over three years, the yields of biochar treatments were higher than those of CK and F treatments. The pod number per plant of the biochar treatment was significantly higher than that of the CK treatment, with the most pronounced effect after the C2 treatment, with pod numbers 21.58%, 33.55%, and 56.40% higher than those after the CK treatment in 2016 to 2018, respectively. The number of grains per plant of C2 was 16.45% and 56.55% higher than that of CK in 2016 and 2018, respectively. In addition, the number of grains per plant of C2 and C3 was higher than that of F for three consecutive years. The 100-grain weight of the C2 treatment was significantly higher than that of CK for three consecutive years, with an average increase of 3.59% per year. The grain weight per plant of CK decreased with increasing cultivation time and was significantly lower in 2018 than in 2016 while the grain weight per plant of the C2 and C3 treatments

increased with cultivation time. From 2016 to 2018, the C2 treatment resulted in a 28.47%, 46.30%, and 77.50% higher grain weight per plant than the CK treatment, respectively (Table 2). In conclusion, biochar application had a cumulative effect on increasing soybean yield, which ensured soybean grain output under continuous cropping.

## 4. Discussion

### 4.1. Effects of Biochar on Soil Properties under Soybean Continuous Cropping

Biochar has rich microporous structure. Moreover, biochar is derived from biomass such as straw; after carbonization, this material has a light weight and low density, and its bulk density is significantly lower than that of the soil [23]. Biochar has a "dilution" effect when applied to the soil, thereby reducing soil compaction and bulk density [24]. These are consistent with our findings (Figure 1). Biochar has a high carbon content and strongly aromatic structure and is highly resistant to physical, chemical, and biological degradation [25]. Therefore, biochar can remain in soil for a long time and perform a sustainable role.

Biochar increased the soil pH under continuous cropping in the present study (Table 1). Studies have shown that there are negatively charged phenols, carboxyl groups, and hydroxyl groups on the surface of biochar, and they can also bind to $H^+$ in soil solution, thereby improving the soil pH [26]. In addition, biochar can adsorb acidic substances secreted by crop roots to the soil, thereby reducing the concentration of acidic substances in the rhizosphere and the pH value of the rhizosphere soil [27], improving the rhizosphere microecological environment under such conditions [28,29]. In this study, the rhizosphere soil pH of the CK and F treatments decreased with the increase in continuous cropping years (Table 1), which was consistent with the results of previous studies [30]. However, there was no significant difference in soil pH between years following the application of biochar in this experiment. This effect probably arose because after a single application of biochar, with the extension of continuous cropping time, the stability or buffering capacity of biochar can protect the soil from acidification due to continuous cropping. It was found in a 10-year experiment that biochar has little effect on soil pH [31]. Therefore, the variation in soil pH with time under biochar application needs further study.

Rhizosphere soil nutrients are the most direct and effective nutrient source for soybean root growth. In this experiment, the biochar significantly increased soil total carbon, nitrogen, phosphorus and potassium, and the rhizosphere soil available nitrogen and potassium were improved to varying degrees with biochar amendments. The promoting effect of biochar on soil nutrients mainly comes from the following aspects (Table 2): Biochar itself is rich in nutrient elements, such as C and other macro- and medium-level trace elements, including N, P, K, S, Ca, Mg, Fe, Zn, Cu, Co, and Mo [25]; thus, it can provide a variety of exogenous nutrients for soil. However, the effective nutrients provided biochar are limited and are mainly available early in the year. More importantly, biochar can uptake and retain soil nutrients, and biochar has a microporous structure and a large specific surface area, which provide a strong adsorption capacity [32], aiding its absorption of nutrients in soil and reducing the loss of nutrients such as nitrogen, phosphorus, and potassium. The aromaticity and surface functional groups of biochar are also beneficial to improving the adsorption capacity of biochar for small molecules, organic matter, and nutrients required for crop growth, such as $NH_4^+$ and $Ca^{2+}$ [33]. In addition, studies have shown that biochar can adsorb water and nutrients that are not used by crops in soil, providing a "storage" effect for soil moisture and nutrients and a certain "slow release" effect; additionally, biochar lessens the migration and transformation of some nutrient elements in soil, reducing the loss of soil nutrients such as nitrogen and phosphorus [34], which improves the soil nutrient level and the effectiveness and availability of soil nutrient elements. The improvement of the rhizosphere soil nutrient level of continuously cropped soybean effectively avoided the negative effects of soil nutrient imbalance and nutrient reduction caused by continuous cropping and ensured more available and sufficient nutrients for promoting soybean growth and development.

*4.2. Effects of Biochar on Soybean Roots under Continuous Cropping*

The beneficial regulation of the physical and chemical properties of rhizosphere soil by biochar created a more favorable soil microecological environment for root growth. Studies found that the application of biochar can increase the root biomass of soybean, significantly promote the total root length and total root surface area of soybean seedlings, and improve root activity [17]; the application of biochar is considered an important management method to increase the root length density and root surface area [16]. The results of this experiment showed that the application of biochar treatments C2 and C3 increased the total root length, total root area and total root volume of continuously cropped soybean (Figure 2), and significantly increased the amount of soybean root exudates (Figure 3), which was basically consistent with the above research results. Relatively speaking, biochar had no significant effect on improving the active absorption area of soybean roots, but this parameter was higher under biochar than under the control (Figure 4). Overall, the application of biochar significantly changed the growth strategy of soybean roots, which was reflected in the morphological and physiological changes. These alterations may help to alleviate or eliminate the inhibitory effect on soybean root growth under continuous cropping; essentially changing the growth state and ability of soybean roots; enhancing the absorption and transport of nutrients, water, and other substances; and promoting the growth and development of soybean. The positive regulatory effects of biochar on roots are closely related to its own structure and characteristics and its regulation of soil physicochemical properties: (1) Soil physical conditions; biochar can reduce soil bulk density, significantly increase soil porosity, and improve soil water, air, heat, and other soil environmental factors, thus creating more physical space and appropriate environmental conditions for root growth and making root growth and development more robust and rapid. (2) Domestic demand for root growth; biochar is rich in nutrients (C, N, P, K, Mg, Ca, S, and others), and the higher pH and other physical and chemical properties of biochar can increase the availability of soil nutrients, which provide direct and effective nutrients for root growth. Biochar provides better soil physical and chemical environmental conditions for root growth, effectively addresses the adverse effects of continuous cropping on the rhizosphere soil microecological environment, and fundamentally changes the root growth strategy.

Additionally, we observed an interesting phenomenon. After the application of biochar, the nodule number, weight, and nitrogenase activity of soybean roots also changed significantly (Figure 5). It is generally believed that nodule growth is greatly affected by nutrient inputs such as exogenous nitrogen. Biochar can increase soil N, P, and K contents and provide suitable N and other nutrient inputs for nodule growth. Previous studies have shown that biochar can increase the amount of soybean nodules under salt stress and increase the absorption and utilization of nitrogen in different parts of soybean plants [35]. In acidic soil, biochar and phosphorus fertilizer can promote soybean nodulation and nitrogen fixation [36]. In the present study, different biochar treatments had different degrees of promoting effects on the number and weight of nodules in soybean roots. The activity of nitrogenase in soybean nodules treated with biochar was increased, which may be related to the application of biochar to improve the soil water content, temperature, and crop photosynthesis [37,38]. The application of biochar improved the physical and chemical characteristics of soil, changed the growth strategy of soybean roots, promoted root growth and nodulation as well as the biological nitrogen fixation ability, effectively inhibited the 'unhealthy' evolution of the soil caused by continuous cropping, and provided an important basis for promoting the healthy and vigorous growth of soybean.

*4.3. Effects of Biochar on Soybean Growth and Yield under Continuous Cropping*

Biochar effectively promoted the growth and development and increased the yield of continuously cropped soybean. Under continuous cropping for 3 years, the soybean yields of C2 and C3 were stable or increased (Table 2), which showed that an appropriate biochar application rate was helpful to achieving stable yields or yield increases under continuous

cropping conditions. In this study, the microporous structure of biochar improved the soil physical structure, and reduced the soil bulk density, indicating good soil physical conditions for continuously cropping. Additionally, biochar contains a large number of nutrients and medium level trace elements needed for crop growth and has a certain absorption and retention effect on soil moisture and nutrients, providing a good soil ecological environment for crop [39]. This effect of biochar was fully confirmed in this experiment; with three consecutive years of biochar application, rhizosphere soil N, P, and K nutrient contents increased to varying degrees (Table 1), effectively reversing the negative effects of soil nutrient deficiency and imbalance caused by continuous soybean cropping. Studies have shown that the application of biochar can enhance microbial activity and accelerate soil physical and chemical reaction processes, such as nitrogen fixation, phosphorus release, and potassium promotion, to increase soil nutrient type, total amount and availability and provide a continuous and effective nutrient supply for soybean growth [23]. Biochar has strong adsorption capabilities, which can inhibit the negative effects of root exudates on soils and roots to a certain extent. Biochar improved the soil structure through its own structure (rich in microporous structure) and further amended the soil physical properties. Meanwhile, biochar influenced the soil chemical properties through its properties (stronger adsorption capacity, and rich in nutrient elements, etc.). Furthermore, the soil microorganisms were inspired by biochar regulating the soil properties, and enhancing the crop root properties, thus improving the soybean growth and productivity. In particular, biochar has a high stability [7], which allows it to exist in the soil for a long time and amend soil properties to improve soybean growth. The positive and sustainable effects of biochar on rhizosphere soil structure, characteristics, root growth, development, and biological nitrogen fixation ability of continuously cropped soybean lay an important foundation for stable yield and yield increases, which is beneficial to fundamentally eliminating the obstacles to continuous cropping soybean production. The underlying mechanisms of these effects are being explored further.

## 5. Conclusions

Under continuous soybean cropping for 3 years, biochar had significant sustainable effects on soil properties (soil bulk density, pH, nutrients) and regulated soybean root growth strategies (morphological development, physiological function, and nodule properties). Ultimately, biochar significantly promoted soybean growth and increased soybean yield ($24 \text{ t·ha}^{-1}$ was the best dose). We discovered that biochar performed a sustainable role in increasing soybean yield by amending soil and root properties to overcome soybean continuous cropping limits, made the soybean continuous cropping possible, and provided a new method for sustainable soybean productivity.

**Author Contributions:** Conceptualization, D.W. and W.Z.; methodology, W.Z.; software, D.W.; validation, D.W. and W.G.; formal analysis, L.X.; investigation, D.W., L.X., W.G., Y.S., Y.W. and H.Z.; resources, W.C.; data curation, D.W.; writing—original draft preparation, D.W.; writing—review and editing, W.Z.; visualization, D.W. and W.Z.; supervision, L.X.; project administration, W.C.; funding acquisition, W.Z. and W.C. All authors have read and agreed to the published version of the manuscript.

**Funding:** This research was funded by the Strategic Priority Research Program of the Chinese Academy of Sciences (XDA28090300), State Key Special Program of Biochar-Fertilizer Technology Research and Industrialization Demonstration (2017YFD0200802-02), State Key Special Program of Biochar-based Fertilizer Development and Application Technology for Soil Fertility Improvement in Rice (2016YFD0300904-4).

**Data Availability Statement:** The data can be obtained from the authors upon request.

**Acknowledgments:** This research was supported the Strategic Priority Research Program of the Chinese Academy of Sciences (XDA28090300), State Key Special Program of Biochar-Fertilizer Technology Research and Industrialization Demonstration (2017YFD0200802-02), State Key Special Program of Biochar-based Fertilizer Development and Application Technology for Soil Fertility Improvement in Rice (2016YFD0300904-4), and the Special Fund for Academicians, and Liaoning Province Major Science and Technology Platform for University (Biochar Engineering and Technical Research Center).

**Conflicts of Interest:** The authors declare no conflict of interest.

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
