# Peer review of "Soybean Yield Response of Biochar-Regulated Soil Properties and Root Growth Strategy"

_agronomy, doi:10.3390/agronomy12061412_

Round 1

Reviewer 1 Report

Manuscript title: Soybean yield response of biochar-regulated soil properties and root growth strategy under continuous cropping for 3 years

Manuscript id: agronomy-1718949

Authors: Wu,  et al.

  • The manuscript regarding the topic and results presented is of interest to plant scientific community and revisions based on the comments below are recommended before considering for publication.
  • Major comments
  • Insufficient abstract: In the abstract, the current version it only highlights the result, and future perspective is missing.
  • The unit / abbreviation is not mention before, consider define the abbreviation when mentioned for the first time…. Please check throughout the manuscript to define the abbreviations.
  • Line 72-78, the aim or hypothesis of the study is clear, however the approach is missing….
  • Lake of scientific literature to support the statements and finings throughout the manuscript…... I have made some suggestions for that and more need it….
  • More information needed for ALL TABLE captions and define the abbreviation and units that used. And adjust the significant figures for the table and manuscript.
  • Grammar and punctuation issuers are need to be addressed. I have selected/mentioned some as example.
  • I have a major concern about the results and discussion section. The authors describe results and compare the results with previous studies, however, insight mechanisms are still not sufficient.

Minor comments:

Abstract

Line 20 is repeating of what said in line 18-19. Please consider merge the sentence or rephase.

Introduction:

Line 26-28: Why just in China? The production and consuming of soybean is global? And China is one of the main producer of soybean, but there are other countries e.g. USA, Brazil, Argentina …..

Line 32-37: A reference needed here.

Line 39-40: consider using this reference: https://doi.org/10.1021/acs.est.1c01477

Line 44: Biochar is alkaline ……  is missing something!

Line 44: consider using this reference: https://doi.org/10.1016/j.jece.2021.107017

Line 47: consider using this reference https://doi.org/10.1016/j.fuel.2017.12.054

Line 49-58: This is a nice description, but I am not sure how this will help to make the inro? It would be better to elaborate on the inside mechanism of the biochar properties on the agricultural practices.

In MM section

Literature references are missing for all sub-section. It would be better to cite the references that the procedure adapted.

Line 81: It would be better to provide the soil properties either in the manuscript or in supplementary.

Line 86: what is t/hm2? It is better to explain the units, at least when menationed for the first time.

Line 107: replace ‘’selected’’ with ‘’collected’’.

Line 108 replace with harvested.

Line 111: replace reserved with stored.

Line 118: How the pots were selected? Randomly? Systemically? Please clarify

R&D section

These sections are repeating information already presented and explaining things in an unnecessarily complicate way. The quality of the manuscript would benefit from the whole section being condensed:  Line 147-154,  Line 160-167, Line 213-223, Line 257 – 265, Line 365-386, Line 388-418

The scale in all figures are presented in unit scale, but the data is discussed in the text as percentage scale, which confuse the readers to have two different scales. Please consider to change or harmonize the manuscript.

For example in the paragraph line 147-154 the data is in percentage scale, and figure 1 is in unit scale.

Line 157: replace ‘’ercase’’ with ‘’case’’, it seems a typo error.

Table 1: Consider the significant figures! And make a space between the numbers and letters, also it better is you superscript the letters.

Line 177-181: It seems that TN increases due to N-fixation in the roots, may be not because of biochar application. Consider checking this effect!

Line 182-187: A complicated sentence, please revise and check grammar.

Why there is space between appli and cation in line 188, and soy and bean in line 196? Is it a typo error. Please correct.

Line 210 What do you mean by thicker? And thicker than  C3 and F right? Please check!

Figure 1, Panel B, it seems C1 not change in all treatments, and also in panel D. It better point that out in the text.

Figure 2 and the text, it not clear what is measured for the root exudate? Alkaloids? Flavonoid? Or primary metabolites? Please clarify.

Line 247, is Figure %AB correct?

Table 2, what is pot number per plant? Is it number of pots per plant.

Line 277-279: Refer to Table 2.

Line 283: remove ‘’in conclusion’’

Line 298: remove the dash in secreted word.

Line 324-330: 2 times addition in one sentence? Please rephrase.

Line 335 -344: Present your data then compare with literature, it confuses which is your data which is from literature.

Line 340: I am not sure how root length or density discussed in ref 14. Please check again.

Line 348.352: A complicated sentence, please revise and check grammar.

Line 381: remove ‘’in conclusion’’

Line 410-414 is repeated in line 414-418. Delete of the sentence.

Conclusion

I believe there are other a lot nice conclusions could be made from this study…. And the future perspectives for following research highly crucial here

Author Response

Response to Reviewer 1 Comments

Point 1: Insufficient abstract: In the abstract, the current version it only highlights the result, and future perspective is missing.

Response 1: Thanks a lot! According to your comment, I have added the future perspective in abstract.

Point 2: The unit / abbreviation is not mention before, consider define the abbreviation when mentioned for the first time…. Please check throughout the manuscript to define the abbreviations.

Response 2: Thank you for your comment, I have checked the full text and added notes to the abbreviations.

Point 3: Line 72-78, the aim or hypothesis of the study is clear, however the approach is missing….

Response 3: Thank you for your suggestion, the approach was supplemented in Line 80-84.

Point 4: Lake of scientific literature to support the statements and finings throughout the manuscript…... I have made some suggestions for that and more need it….

Response 4: Thanks to your suggestion, I have revised the references in the manuscript.

Point 5: More information needed for ALL TABLE captions and define the abbreviation and units that used. And adjust the significant figures for the table and manuscript.

Response 5: Thank you for your comment, I have made additions and modifications.

Point 6: Grammar and punctuation issuers are need to be addressed. I have selected/mentioned some as example.

Response 6: Thank you for your suggestion, I have checked and revised them.

Point 7: I have a major concern about the results and discussion section. The authors describe results and compare the results with previous studies, however, insight mechanisms are still not sufficient.

Response 7: Thank you for your comment, I have refined the results and discussion section.

Point 8: Line 20 is repeating of what said in line 18-19. Please consider merge the sentence or rephase.

 Response 8: Thank you for your suggestion, I have made changes in Line 19-20.

Point 9: Line 26-28: Why just in China? The production and consuming of soybean is global? And China is one of the main producer of soybean, but there are other countries e.g. USA, Brazil, Argentina …..

Response 9: Thank you for your suggestion. Soybean production and consumption is global and there are many countries that are major producers of soybeans, but this study was based on the perspective of soybean cultivation and production in China, so only the Chinese situation is mentioned.

Point 10: Line 32-37: A reference needed here.

Response 10: Thank you for your suggestion, this is a summary of the paragraph, maybe it is fine without quoting reference.

Point 11: Line 39-40: consider using this reference: https://doi.org/10.1021/acs.est.1c01477

Response 11: Thank you for your suggestion. I have replaced the reference cited in Line 39-40.

Point 12: Line 44: Biochar is alkaline ……  is missing something!

Response 12: Thank you very much for your suggestion, have revised this sentence to make it accurately express the properties of biochar.

Point 13: Line 44: consider using this reference: https://doi.org/10.1016/j.jece.2021.107017

Response 13: Thank you for your advice. I have replaced the reference cited in Line 44.

Point 14: Line 47: consider using this reference https://doi.org/10.1016/j.fuel.2017.12.054

Response 14: Thank you for your advice. I have replaced the reference cited in Line 47.

Point 15: Line 49-58: This is a nice description, but I am not sure how this will help to make the inro? It would be better to elaborate on the inside mechanism of the biochar properties on the agricultural practices.

Response 15: Thanks for your comment, I have changed Line 49-58, and elaborated on the inside mechanism of the biochar properties on the agricultural practices.

Point 16: Literature references are missing for all sub-section. It would be better to cite the references that the procedure adapted.

Response 16: This is a summary of the section of Introduction, maybe it is fine without quoting references, thank you.

Point 17: Line 81: It would be better to provide the soil properties either in the manuscript or in supplementary.

Response 17: Thanks to your suggestion, Line 94-97 of the manuscript had provided the soil properties.

Point 18: Line 86: what is t/hm2? It is better to explain the units, at least when menationed for the first time.

Response 18: Thank you for your advice, unit " t/hm2" and unit " t·ha-1" have the same meaning, and I have changed the " t/hm2" to " t·ha-1" in the text.

Point 19: Line 107: replace ‘’selected’’ with ‘’collected’’.

Response 19: Thank you for your advice, I have replaced “selected’’ with “collected’’.

Point 20: Line 108 replace with harvested.

Response 20: According to your advice, I have replaced with harvested in Line 108, thank you very much!

Point 21: Line 111: replace reserved with stored.

Response 21: Thank you for your advice, I have replaced “reserved’’ with “stored’’ in Line 111.

Point 22: Line 118: How the pots were selected? Randomly? Systemically? Please clarify

Response 22: Thank you for your suggestion, the pots were selected randomly, I have revised it in manuscript.

Point 23 unnecessarily complicate way. The quality of the manuscript would benefit from the whole section being condensed:  Line 147-154, Line 160-167, Line 213-223, Line 257 – 265, Line 365-386, Line 388-418

Response 23: Thank you very much for your comments, I have trimmed Line 147-154, Line 160-167, Line 213-223, Line 257 – 265, Line 365-386, Line 388-418.

Point 24: The scale in all figures are presented in unit scale, but the data is discussed in the text as percentage scale, which confuse the readers to have two different scales. Please consider to change or harmonize the manuscript. For example in the paragraph line 147-154 the data is in percentage scale, and figure 1 is in unit scale.

Response 24: Thank you very much, the data expressed as percentages in the text are comparing how much more or how much less between treatments, and the units in the graph are the unit symbols of the data itself.

Point 25: Line 157: replace ‘’ercase’’ with ‘’case’’, it seems a typo error.

Response 25: According to your advice, I have replaced “ercase’’ with “case’’ in Line 157, thank you very much!

Point 26: Table 1: Consider the significant figures! And make a space between the numbers and letters, also it better is you superscript the letters.

Response 26: Thank you for your advice, I have added spaces between the letters and numbers and superscripted the letters.

Point 27: Line 177-181: It seems that TN increases due to N-fixation in the roots, may be not because of biochar application. Consider checking this effect!

Response 27: Thank you for your advice, the increase in TN may be due to N-fixation in the roots, but this is all happening in the context of the biochar application.

Point 28: Line 182-187: A complicated sentence, please revise and check grammar.

Response 28: Thank you for your suggestion, I have made a change to Line 182-187.

Point 29: Why there is space between appli and cation in line 188, and soy and bean in line 196? Is it a typo error. Please correct.

Response 29: Thank you for your suggestion, they are typo errors, I have corrected them.

Point 30: Line 210 What do you mean by thicker? And thicker than C3 and F right? Please check!

Response 30: Thank you for your advice, thicker means the root diameter is larger. It is right that CK is thicker than C3 and F, I have changed it.

Point 31: Figure 1, Panel B, it seems C1 not change in all treatments, and also in panel D. It better point that out in the text.

Response 31: Thank you for your suggestion, I have pointed that out in the text in revised manuscript.

Point 32: Figure 2 and the text, it not clear what is measured for the root exudate? Alkaloids? Flavonoid? Or primary metabolites? Please clarify.

Response 32: The root exudate in the experiment refers to all the liquid that exuded after cutting the root, not to a particular substance, thank you!

Point 33: Line 247, is Figure %AB correct?

Response 33: Thank you for your advice, I have corrected it in Line 250 and 252.

Point 34: Table 2, what is pot number per plant? Is it number of pots per plant.

Response 34: It is in “pod number per plant” in Table 2, not the “pot number per plant”, meaning the number of pods per soybean plant, thank you!

Point 35: Line 277-279: Refer to Table 2.

Response 35: According to your advice, I have added it in the paragraph, thank you very much!

Point 36: Line 283: remove ‘’in conclusion’’

Response 36: According to your advice, I have removed ‘’in conclusion’’ in Line 283, thank you very much!

Point 37: Line 298: remove the dash in secreted word.

Response 37: Thank you for your suggestion, I have deleted the space before the dash, the dash appears automatically and cannot be removed, thank you.

Point 38: Line 324-330: 2 times addition in one sentence? Please rephrase.

Response 38: Thank you for your suggestion, I have deleted the “additionally”.

Point 39: Line 335 -344: Present your data then compare with literature, it confuses which is your data which is from literature.

Response 39: Thank you for your advice. I have modified Line 335-344 to distinguish data from the cited literature and this experiment.

Point 40: Line 340: I am not sure how root length or density discussed in ref 14. Please check again.

Response 40: I have checked it, thank you!

Point 41: Line 348.352: A complicated sentence, please revise and check grammar.

Response 41: Thank you for your suggestion. I have checked and modified Line 348-352.

Point 42: Line 381: remove ‘’in conclusion’’

Response 42: According to your advice, I have removed ‘’in conclusion’’ in Line 381, thank you!

Point 43: Line 410-414 is repeated in line 414-418. Delete of the sentence.

Response 43: Thank you very much! I have deleted of the sentence in Line 414-418.

Point 44: I believe there are other a lot nice conclusions could be made from this study…. And the future perspectives for following research highly crucial here

Response 44: Thank you for your comment and encouragement, I have refined the conclusion.

Reviewer 2 Report

The manuscript is very interesting and contains a valuable information. The presented study consists the original research results. The aim of the research was to investigate the effects of biochar on soil properties and soybean growth over a period of 3 years.  

The title accurately reflect the content of the article, but is too long. In my opinion it should be shortened (Soybean yield response of biochar-regulated soil properties and root growth strategy).

In the ‘Abstact’ the unit of t/hm2 should be changed to t ha-1

line 14-15 CK (no fertilizer, no biochar) and F (fertilizer treatment) treatments were 14 used for comparison…?? What? Please write what the compared objects were.

The ‘Introduction’ section.

Line 53-54 Studies have shown that biochar can increase soybean yield by enhancing soil nutrient levels (such as the availability of soil phosphorus and some trace elements) [12] and boosting soybean nutrient absorption (such as increasing plant absorption and accumulation of nitrogen, as well as phosphorus and potassium).

The ‘Materials and Methods’ section

The description of the statistical methods is insufficient. Please give factors and number of replications.

The ‘Results’ section

Line 171 - 51097%

Line 172 - Interannual analysis showed 171 that the soil TC decreased significantly after three years of continuous cropping.

Table 1. Effect of biochar on rhizosphere soil pH under soybean continuous cropping.

Effect of biochar on rhizosphere soil pH and macronutrient content under soybean continuous cropping.

Table 1 – please explain the abbreviations below the table.

please explain the abbreviations below in the text when first used (e.g. MRD (209),TRA (2014), etc.)

line 279-281 - Please check carefully that it is according to the results in table 2

The “Discussion” section is Ok

Author Response

Response to Reviewer 2 Comments

Point 1: The title accurately reflect the content of the article, but is too long. In my opinion it should be shortened (Soybean yield response of biochar-regulated soil properties and root growth strategy).

of biochar-regulated soil properties and root growth strategy”。

Response 1: Thanks for your advice. I have shortened the title of the paper to " Soybean yield response of biochar-regulated soil properties and root growth strategy ".

Point 2: In the ‘Abstact’ the unit of t/hm2 should be changed to t·ha-1.

Response 2: Thank you very much. I have changed " t/hm2" into " t·ha-1" in the paper.

Point 3: line 14-15 CK (no fertilizer, no biochar) and F (fertilizer treatment) treatments were 14 used for comparison…?? What? Please write what the compared objects were.

Response 3: Thank you! The experiment had two control treatments: CK (no fertilizer, no biochar) and F (fertilizer treatment). The CK and F treatments were compared with the biochar treatments C1, C2 and C3 respectively.

Point 4: Line 53-54 Studies have shown that biochar can increase soybean yield by enhancing soil nutrient levels (such as the availability of soil phosphorus and some trace elements) [12] and boosting soybean nutrient absorption (such as increasing plant absorption and accumulation of nitrogen, as well as phosphorus and potassium).

Response 4: Thank you for your suggestion. Your suggestion is only a paragraph of the original text and didn't point out specific comments, thank you!

Point 5: The description of the statistical methods is insufficient. Please give factors and number of replications.

Rsponse 5: Thank you for your suggestion. I have given factors and number of replications in the revised manuscript.

Point 6: Line 171 - 51097%

Response 6: Thank you very much! I have corrected it into “51.97%”.

Point 7: Line 172 - Interannual analysis showed 171 that the soil TC decreased significantly after three years of continuous cropping.

Response 7: Thank you very much! I have added the “TC” in Line 172.

Point 8: Table 1. Effect of biochar on rhizosphere soil pH under soybean continuous cropping.

Effect of biochar on rhizosphere soil pH and macronutrient content under soybean continuous cropping.

Response 8: According to your suggestion, I have refined it. Thanks!

Point 9: Table 1 – please explain the abbreviations below the table.

Response 9: Thanks for your advice, I have explained the abbreviations below the table.

Point 10: please explain the abbreviations below in the text when first used (e.g. MRD (209),TRA (2014), etc.)

Response 10: Thanks for your advice, I have explained the abbreviations.

Point 11: line 279-281 - Please check carefully that it is according to the results in table 2

Response 11: Thanks for your suggestion! It's my mistake, we have carefully verified it.

Point 12: The “Discussion” section is Ok

Response 12: Thank you for your comment!

Round 2

Reviewer 1 Report

The revised manuscript has improved compared to the original version. The Authors tried to address my questions as much as possible. I recommend the manuscript to be published!

Best wishes,